# Determinants of implementing deprescribing for older adults in English care homes: a qualitative interview study

Krystal Warmoth [1,2] Jessica Rees [3] Jo Day,[4,5] Emma Cockcroft,[4,5] Alex Aylward,[6] Lucy Pollock,[7] George Coxon,[8] Trudy Craig,[9] Bridget Walton,[5] Ken Stein[4,5]

For numbered affiliations see end of article.

**Correspondence to**
Dr Krystal Warmoth;
k.warmoth@herts.ac.uk

## ABSTRACT

**Objectives** To explore the factors that may help or hinder deprescribing practice for older people within care homes.

**Design** Qualitative semistructured interviews using framework analysis informed by the Consolidated Framework for Implementation Research (CFIR).

**Setting** Participants were recruited from two care home provider organisations (a smaller independently owned organisation and a large organisation) in England.

**Participants** A sample of 23 care home staff, 8 residents, 4 family members and 1 general practitioner were associated with 15 care homes.

**Results** Participants discussed their experiences and perceptions of implementing deprescribing within care homes. Major themes of (1) deprescribing as a complex process and (2) internal and external contextual factors influencing deprescribing practice (such as beliefs, abilities and relationships) were interrelated and spanned several CFIR constructs and domains. The quality of local relationships with and support from healthcare professionals were considered more crucial factors than the type of care home management structure.

**Conclusions** Several influencing social and contextual factors need to be considered for implementing deprescribing for older adults in care homes. Additional training, tools, support and opportunities need to be made available to care home staff, so they can feel confident and able to question or raise concerns about medicines with prescribers. Further work is warranted to design and adopt a deprescribing approach which addresses these determinants to ensure successful implementation.

## INTRODUCTION

Older people living in UK care homes often experience polypharmacy, which is commonly defined as receiving five or more concurrent medicines a day.[1] In England, the 2021 National Overprescribing Review found older people to be at greater risk from polypharmacy; over half of the people over the age of 80 take eight or more medicines a day,[2] many of whom live in care homes. Polypharmacy is widespread, with over 60% of residents taking

### STRENGTHS AND LIMITATIONS OF THIS STUDY

⇒ Data collection and analysis were informed by a comprehensive, well-recognised implementation science framework.

⇒ This research adopted a strong patient and public involvement and partnership approach.

⇒ The number of respondents who participated during the pandemic and the high demand for healthcare services should be acknowledged.

⇒ The majority of participants were care home staff, so the findings reflect mostly their perspectives and experiences.

⇒ Methods were modified from the original protocol due to the COVID-19 pandemic.

five or more medicines,[3] and increasing with system-wide overprescribing.[2 4] Some older adults are prescribed multiple medicines that are unlikely to improve clinical outcomes, are clinically unnecessary or may lead to harm.[5–8] One-half of care home residents are exposed to potentially inappropriate medicines.[9] Care home staff, residents and family members stress the high prevalence, fears about the health and safety consequences and burden of polypharmacy in care homes.[10]

Reducing or stopping prescription medicines which may no longer be providing benefit or where the harms outweigh the benefits, known as deprescribing,[11] can mitigate these harms and be safe.[12–15] The National Institute for Health and Care Excellence (NICE) recommends deprescribing as part of the comprehensive medication review of a person with multiple long-term conditions.[16–18] While appropriate deprescribing is usually commended and may be cautiously undertaken to good effect,[15] there is a lack of information about how to implement it safely and appropriately.[19] Recommendations overlook the specific contextual factors and

various stakeholder views of care homes.[20] Deprescribing in this setting can be challenging due to the different concerns of residents, staff, clinicians and family members and differences in care home structures.

Previous deprescribing research explored generic views towards barriers and facilitators to deprescribing[21–27] and only focused on the perspectives of a single stakeholder group, such as patients[25 27] or general practitioners (GPs)[22–24], and rarely included crucial care home staff views.[28] Different stakeholder groups from long-term care facilities prioritise different factors.[29] There is a need for a better understanding of the factors influencing how deprescribing is implemented, considering different stakeholders.

A recent summary of the perceptions of deprescribing in long-term care facilities from nine different countries found social influences and environmental factors were perceived as the key barriers and enablers.[30] Considering these findings, the present study investigated the process of implementing deprescribing and these contextual factors in English care homes, including the roles and relationships of different stakeholders, which influence it. Furthermore, implementation activities of deprescribing in care homes are typically not well described and poorly understood.[19 31] A better understanding of how contextual factors facilitate or hinder deprescribing, informed by an implementation framework, is needed to support the translation of deprescribing recommendations into practice.[32 33]

The present study presents the findings of the first work package of the STOPPING project (for the original protocol, see Warmoth *et al*[34]). The overall aim was to investigate the factors which influence deprescribing for older adults in care homes. The specific objectives of the research were to:

1. Identify the factors which influence deprescribing for older adults in care homes.
2. Explore factors influencing deprescribing across a variety of stakeholders and care home provider organisations.
3. Use the Consolidated Framework for Implementation Research (CFIR) to help understand the possible determinants of implementing deprescribing for older adults in a care home setting.

## METHODS
### Design
We used qualitative methods and adopted a pragmatic approach. Originally, focus groups with care home staff and healthcare professionals and in-person observations of care homes were planned, but were no longer feasible due to the COVID-19 pandemic. The original methods were changed to allow for remote data collection. Individual or dyadic interviews were conducted with all participants via Microsoft Teams, Zoom or by telephone.

We gathered and compared perspectives from various stakeholders from two different types of care home provider organisations and used an implementation science framework to inform the data collection and analysis. The CFIR was used as an overarching framework to ensure that implementation was central to data collection and analysis.[35] CFIR (2009 version) is a well-established, theoretically based implementation science framework, comprised of 39 constructs divided into 5 domains.[35] CFIR focuses on identifying and understanding constructs that can shape the implementation and the routinisation of health services,[36] making it appropriate for care homes.

### Setting
The two care home providers were selected for this research as they represent contrasting models of care home provision. That is, one is a smaller independently owned organisation consisting of two residential care home sites and a large organisation with more than 25 care homes in several locations across England. They were also selected as they mainly care for older adults (over 65 years old). Care home sites included in this study were selected in partnership between the research team and senior care home staff.

### Participants
Recruitment and data collection occurred from December 2021 to May 2022. See table 1 for participant details and table 2 for the care home details. Care home and other healthcare staff were eligible for the study if they were currently working directly with older adults with polypharmacy and/or in the care home setting and could converse in English without an interpreter or professional assistance. Participants were purposefully recruited from distinct types of care home providers using established networks of care home organisations. Care home staff were selected and approached by the regional managers or directors of the care home organisations, who were known to the researchers.

Care home staff approached residents and their family members about taking part in the study. The eligibility criteria for care home residents taking part in the research were being a resident of the participating care home, aged 65 and over, taking multiple medications or having experience of polypharmacy, ability to converse in English without an interpreter or professional assistance, and having an absence of serious cognitive impairment, as identified by the care home staff or healthcare profession, and have capacity to consent to participation. Care home residents' family members or friends were eligible if their resident was taking multiple medications at the care home participating in the study and ability to converse in English without an interpreter or professional assistance. Residents with severe cognitive impairment which inhibits consent were excluded, but their family members and carers were included, as this group are particularly at risk of overall drug burden and often benefit from deprescribing interventions.

**Table 1** Participant characteristics (n=36)

| | n (%) | M (SD) |
|---|---|---|
| Care home staff and healthcare professionals (n=24) | | |
| Age | | 44.65 (12.04) |
| Gender | | |
| Female | 21 (87.50) | |
| Male | 3 (12.50) | |
| Ethnic group | | |
| White | 23 (95.80) | |
| Asian | 1 (4.20) | |
| Education | | |
| General certificate of education | 3 (12.50) | |
| Undergraduate degree | 5 (20.8 0) | |
| National vocational qualification | 14 (58.30) | |
| Postgraduate degree | 2 (8.30) | |
| Employment duration (in months) | | 47.83 (59.27) |
| Key responsibilities and duties | | |
| Assistance with daily tasks in care home | 14 (58.33) | |
| Administering medicines | 17 (70.83) | |
| Providing company and assistance in leisure activities | 5 (20.83) | |
| Developing care plans | 20 (83.33) | |
| Planning/overseeing work of other staff members | 21 (87.50) | |
| Prescribing and medicine review | 16 (66.67) | |
| Other | 14 (58.33) | |
| Hours worked per week | | |
| Less than 32 hours | 3 (12.50) | |
| More than 32 hours | 21 (87.50) | |
| Family carers (n=4) | | |
| Age | | 64.75 (1.26) |
| Gender | | |
| Female | 3 (75.00) | |
| Male | 1 (25.00) | |
| Ethnic group—white | 4 (100.00) | |
| Relationship to resident | | |
| Son | 1 (25.00) | |
| Daughter | 2 (50.00) | |
| Sibling | 1 (25.00) | |
| Residents (n=8) | | |
| Age | | 88.75 (5.06) |
| Gender | | |
| Female | 6 (75.00) | |
| Male | 2 (25.00) | |
| Ethnic group—white | 8 (100.00) | |
| Perceived health status | | 3.50 (0.93) |
| Excellent | 1 (12.50) | |
| Very good | 0 | |
| Good | 3 (37.50) | |

Continued

**Table 1** Continued

| | n (%) | M (SD) |
|---|---|---|
| Fair | 3 (37.50) | |
| Poor | 1 (12.50) | |
| Number medicines daily | | 7.71 (3.99) |
| Number of conditions | | 6.40 (2.30) |

One participant could not remember the number of medicines they took daily, so they were not included in the analysis for that item.

### Data collection

Demographic data were collected from all participants following the interview with a brief survey. Semistructured interviews were conducted with residents, their family members, care home staff and healthcare professionals about their experiences and beliefs to identify the factors influencing deprescribing and the current deprescribing practice in different care home settings. Interview topic guides were informed by CFIR constructs identified in the previous literature.[21 25–27] See online supplemental file 1 for the interview guides. Interviews were conducted by a qualified female researcher with more than 10 years of experience (KW) and audio recorded and professionally transcribed verbatim for analysis. Reflective notes were taken to supplement the transcripts. NVivo V.12 was used to manage the data.

Qualitative data collection was conducted until data saturation occurred, meaning that no new ideas were being generated. The sample size was determined using previous literature on qualitative methods and using purposeful sampling. We estimated the minimum sample size of 24 was needed, based on work that found data saturation and variability to be present as early as 6 interviews.[37]

### Data analysis

A framework analysis approach[38] was employed for data analysis. A CFIR-informed codebook was adapted to address the research question and to consider distinct levels of analysis (individual, organisational and community); see online supplemental file 2. Transcripts were deductively coded within the domains and constructs of CFIR by a single researcher (JR). Then, researchers (KW and JR) developed the major themes across the domains inductively. Discussions with patient and public involvement (PPI) representatives helped to refine the major themes and the implications of the research.

### Patient and public involvement (PPI)

This research adopted a strong PPI and partnership approach with residents, staff and family carers. Before data collection, a PPI workshop was held with care home residents and staff at one care home. This workshop informed the production of study materials and ensured they were appropriate for and understandable by care home residents and staff. A pilot interview with an independent member of a local PPI group was conducted,

**Table 2** Care home characteristics

| | Participants, n | Beds, n | Occupancy (%) | Total number of staff | Dementia specialty | Nursing care |
|---|---|---|---|---|---|---|
| **Large care home provider** | | | | | | |
| Care home 1 | 2 residents, 1 staff | 60 | 88 | 85 | Yes | No |
| Care home 2 | 2 family | 90 | 89 | 140 | Yes | Yes |
| Care home 3 | 2 staff, 1 family | 55 | 100 | 75 | Yes | Yes |
| Care home 4 | 2 staff | 34 | 75 | 36 | No | No |
| Care home 5 | 1 staff, 1 resident, 1 family | 54 | 81 | 57 | No | No |
| Care home 6 | 1 staff, 1 resident | 50 | 76 | 32 | Yes | No |
| Care home 7 | 1 staff | 68 | 83 | 74 | Yes | No |
| Care home 8 | 2 staff | 46 | 85 | 62 | Yes | No |
| Care home 9 | 1 staff | 44 | 68 | 36 | Yes | No |
| Care home 10 | 3 staff, 2 residents | 37 | 91 | 54 | Yes | No |
| Care home 11 | 1 staff, 1 resident | 40 | 95 | 49 | No | No |
| Care home 12 | 2 staff | 38 | 60 | 42 | No | No |
| Care home 13 | 1 staff | 39 | 79 | 50 | Yes | No |
| **Small independent owned provider** | | | | | | |
| Care home 14 | 3 staff, 1 resident | 17 | 100 | 25 | Yes | No |
| Care home 15 | 2 staff | 19 | 100 | 26 | Yes | No |

and as a result, opportunities for breaks were included and minor changes to the wording of the topic guides were made.

Researchers (JR and KW) and a PPI representative (AA) met regularly to discuss the analysis plan, review and refine the codebook, develop the themes and interpret the findings.

A final PPI workshop was held to discuss the findings and refine the dissemination strategy with a group of family carers and care home workers. Two researchers and the PPI representative presented the findings and facilitated group discussions about the interpretation of the findings and dissemination. Attendees stated how they shared similar experiences with participants and the varied procedures in care homes.

## RESULTS

In total, 36 interviews were conducted with 23 care home staff, 8 residents, 4 family members and 1 GP who were associated with 15 care homes. Interviews lasted on average approximately 41 min (ranging from 16 to 94 min). Two major themes were developed: (1) deprescribing as a complex process and (2) internal and external contextual factors influencing deprescribing in care homes. These themes were interrelated and spanned several CFIR constructs and domains. See table 3 for the major themes, CFIR determinants and supporting quotes.

### Deprescribing as a complex process

This theme related to the activities and strategies described to implement deprescribing in a care home setting, relating to the CFIR domain of process. Participants discussed deprescribing as a complex process with multiple steps. It included preparation and planning, involvement of multiple people with different roles and ongoing monitoring and evaluation.

### Preparation and planning deprescribing

Conversations about deprescribing could be initiated by a regular medication review, an observed change in the resident or the resident's preference. Before stopping or reducing medications, care home residents and staff discussed that '*health checks*' and comprehensive medication reviews should be conducted. Care home staff and the GP expressed how, during a medicine review, there was a need to fully understand the resident's medications and the rationale for each one's use. If a decision was made to deprescribe then the care home needed to know how to do it (ie, tapering over time or stopping straight away). Care home staff discussed supporting resident involvement and its difficulties with some residents (eg, cognitive impairment). They considered that it was important to explain any changes and review how the resident was feeling on certain medications, before stopping or reducing medications.

**Table 3** Summary of factors influencing the implementation of deprescribing in care homes from interviews with residents, family carers, care home staff and healthcare professionals

| Major themes | CFIR domains | CFIR constructs | Definitions | Supporting quotes |
|---|---|---|---|---|
| Deprescribing as a process | Process | Planning | Degree to which tasks or behaviours for implementing deprescribing are developed in advance. | "I think there'd be a bit of a process involved. I think probably the family members and staff, the people that know the individuals well, or better than anybody else, they'd have to have an input for their opinion. And then the person that prescribes the medicines, would have to review it and decide what they think for the best." (Care home 14, care assistant) |
| | | | | "The pharmacist poking his nose in may be good, or it may not be, I don't know." (Care home 14, resident) |
| | Engaging | External change agents | Individuals who are affiliated with organisations/services outside the care home who formally influence or facilitate deprescribing. | "So I think they're essential for coordinating everyone, really, and actually they're better at doing that than the GP will ever be." (Care home 8, GP) |
| | | Key stakeholders | Care home staff roles that influence deprescribing. | |
| | | Innovation participants—resident | Resident role in deprescribing (including how needs and preferences and included in decisions, and engagement and involvement examples/challenges). | "The resident themselves; they should have an input if they have capacity to do it, if they're able to retain the information about their medication." (Care home 12, deputy manager) |
| | | Innovation participants—family carer | Family carer role in deprescribing (including how needs and preferences and included in decisions, and engagement and involvement examples/challenges). | "And if they lack capacity, even if they don't lack capacity, then their family, close family, next to kin, power of attorney, we would ask and consult them as well, for anything that changes with any of their care." (Care home 6, deputy manager and medication lead) |
| | Reflecting and evaluating | | Feedback about the progress and quality of deprescribing and monitoring following deprescribing. | "Well, I suppose a final review, at some point, so that's predetermined and there should be a point where it's agreed that there should be a review of how it was effective, but whether it's worked, I suppose." (Care home 2, family carer) |
| Internal and external contextual factors | Individual characteristics | Knowledge and beliefs about deprescribing—resident | Resident attitudes and opinion towards deprescribing. | "I think I would have concerns, certainly, with some medication you have to be careful when you come off with the after-effects. And it has to be done in carefully? I wouldn't just accept everything, you have an understanding why." (Care home 1, resident) |
| | | Knowledge and beliefs about deprescribing—care home staff | Care home staff attitudes and opinion towards deprescribing. | "And they don't necessarily need it, and they come up with it, and we've trialled them coming off of it and they're absolutely fine. And it's nicer for them not being pumped full of medication, rattling around daily. If it's not needed, and they've come here to live out the rest of their life, who really wants to be taking lots of medication every day?" (Care home 12, supervisor) |
| | | Self-efficacy—resident | Resident belief in their own knowledge, capabilities and ability to action deprescribing. | "Well, if the medical professional thought it was all right, I would just take their word for it...I think we would take the advice of the doctor." (Care home 1, resident) |
| | | Self-efficacy—care home staff | Care home staff belief in their own knowledge, capabilities and ability to action deprescribing. | "I think we're able to certainly speak up and challenge. I don't think we'd start questioning and going, they shouldn't be on this, but we would certainly raise concerns and making sure that people are having regular medication reviews, so that they're not on anything unnecessarily." (Care home 4, deputy manager) |

Continued

**Table 3** Continued

| Major themes | CFIR domains | CFIR constructs | Definitions | Supporting quotes |
|---|---|---|---|---|
| | Outer setting | Cosmopolitanism | The degree to which the care home is networked with other external organisations. | "Of course, it's interdependent, because they depend on each other… it depends on the advice of the carer giving to the doctors about what's happening to a patient. And then the doctors summing it up and advising that they should have a certain medication specifically for that patient, or that resident." (Care home 6, resident) |
| | Inner setting | Access to knowledge and information | Ease of access to digestible information and knowledge about deprescribing and medicines and how to incorporate it into the way staff provide care for residents. | "We just get told, oh, they've had a blood test, they need to have this medication. And it's like, but what did the blood tests show? And it's like they don't always tell us, and it's just like, really? Give us more information, please!" (Care home 9, deputy manager) |
| | | Available resources | The level of resources the care home dedicated for making changes to the way things are done and ongoing operations including physical space and time. | "I think the main thing we need, is time. If I think of the care home I look after, there's 65 residents, so I've got to find… If I want to do it properly, I'd have find at least 65 hours a year to do it regularly, and follow-ups. So it's really quite hard to manage…So time is the biggest problem, and also, and time for care home staff. I mean, they're as pressured as anyone else, for them to actually spend… Take an hour of their day out to talk about each one of their residents, again, is quite onerous on them." (Care home 8, GP) |
| | | Networks and communication | The nature and quality of webs of social networks, and the nature and quality of formal and informal communications within the care home. | "Because, obviously, it's on the [Medicine Administration Record] sheet, it's crossed off on our [Medicine Administration Record] sheets, stopped by the GP. It's documented in their care plans. It's documented on our [Patient Centred Software]. It's all well-documented, and like you do medication, you have to follow your [Medicine Administration Record] sheets, so it's always there so everyone can see what's stopped and why, and who stopped it and what date." (Care home 15, head of care) |
| | | Tension for change | The degree to which the current situation as intolerable or needing change. | "So, yeah, I definitely think that they need to be reviewed regularly, because you don't want people taking medication that they no longer need, that they no longer require, if that makes sense?" (Care home 4, supervisor) |
| | | Resident needs and resources | The extent to which the needs of care home residents, as well as barriers and facilitators to meet those needs, are accurately known and prioritised by the care home. | "So it comes about from us knowing our residents, and being able to make sure we pass that information clearly and accurately and timely onto the surgery, as I mentioned before." (Care home 4, deputy manager) |
| | Characteristics of deprescribing (intervention) | Evidence strength and quality | Perceptions of the quality and validity of evidence supporting the belief that deprescribing will have positive outcomes for residents or will do no harm (adverse outcomes). | "I'm aware there is some, but I can't quote anything offhand. We get sent it by our medicines management team quite often, about reviewing. So particularly with antipsychotics…But, yeah, so there is some, and there is our medicines management team at CCG who are really good about supporting us with evidence to things." (Care home 8, GP) |
| | | Cost | Costs of deprescribing and costs associated with implementing deprescribing including investment, supply and opportunity costs. | "I don't think there's any direct cost involved to us, apart from our time. But it would be a nice cost to the resident, because, hopefully, we'd be able to spend more quality time with them, rather than being shove that down your throat, we've got another 40 more to do." (Care home 11, manager) |

Continued

**Table 3** Continued

| Major themes | CFIR domains | CFIR constructs | Definitions | Supporting quotes |
|---|---|---|---|---|
| | | Adaptability | The degree to which deprescribing can be adapted, tailored, refined or reinvented to meet needs at organisation/provider (care home) and individual (resident) level. | "Well, yeah, we're all slightly different. We think we're the same, but we're not and our bodies react differently. But what works – if I come back to this, all these tablets, to my knowledge, work for me." (Care home 14, resident) |
| | | Complexity | Perceived difficulty of the deprescribing. | "It's looking for any changes. It's much easier to do this with people who've got capacity, who can tell you. With people with dementia whose communication is limited, you do have to be monitoring more, actually. What are they – are they normal for them?" (Care home 2, nurse supervisor) |
| | | Relative advantage | Perception of the outcome deprescribing (ie, better, worse, no change) versus an alternative solution (ie, continuing medications). | "I don't want her to take things that don't do her - don't do anything." (Care home 2, family carer) |
| | | Trialability | The ability to test deprescribing on a small scale in the care home, and to be able to reverse course (undo implementation) if warranted (ie, restart medications). | "Well, if it was clear to the medical people, if they wouldn't be, if it was clear to them, and they could go communicate to me, they would try and I'd give it a try, as long as I could go back if I wasn't happy." (Care home 1, resident) |

GP, general practitioner.

## Engaging multiple people

For deprescribing to happen, all participants discussed how it involved the engagement of various individuals, internal and external to the care homes. Each had a specific role and expertise, which was highlighted by participants. Care home staff, residents and their families expressed how care home staff are residents' advocates and/or intermediaries and often '*know them best*' (ie, the resident's individual needs and resources); whereas GPs made the ultimate decisions about if and how deprescribing should happen. A resident discussed the distinct roles and how they were '*interdependent because they depend on each other*'. The involvement of these different people and their respective knowledge and beliefs could help and hinder deprescribing. For example, family members were discussed by care home staff as either encouraging or objecting to the resident's medicines being deprescribed.

The role of a pharmacist in the deprescribing process was more ambiguous. Some residents and care home staff thought the role of a pharmacist was only dispensing medication and, hence, were not involved in any aspect of deprescribing or unsure how they could help. Conversely, if a care home had experienced additional support from pharmacists, such as leading medicine reviews or as a resource for questions about medicines, they were more likely to report how a pharmacist's input was valuable to deprescribing. Pharmacists were considered experts in medicines, but the ultimate decision is made by the GP unless the resident had secondary care where another clinician makes prescribing decisions. Despite the knowledge and expertise of care home staff about the resident and the pharmacist about medicines, all participants (especially, residents) expressed how deprescribing decisions would be done by the GP. Occasionally, other healthcare professionals (eg, the mental health team, geriatricians or specific condition nurses) were mentioned but they were often described as being consulted by the GP to make their decisions.

## Monitoring and evaluation

Another key part of the deprescribing process that participants discussed was the monitoring and evaluation after medication changes. This was a consistent theme across all participant groups, but especially the care home staff as they were the ones observing residents following medication changes. Care home staff emphasised how important it was for them to know about medications to make relevant observations regarding any benefits or side effects. The types of observations required depend on which medications were changed (eg, blood pressure, behaviour, alertness and appetite). The timings and workload required for feedback also varied depending on the complexity of the resident's health and the types of medication changes. Care home staff and the GP discussed how the effects of stopping or reducing certain medications may take longer than others, thus impacting the monitoring, feedback and review process. This monitoring was described as actively ongoing to optimise medications

for residents and included the possibility of restarting medications.

## Internal and external contextual factors

This theme concerns the contextual factors, including beliefs, abilities and relationships, reported to influence deprescribing in a care home setting. These CFIR determinants are related to the characteristics of staff and residents (views about deprescribing benefits, ability to question and contribute) as well as the relationships and communication between care homes and the healthcare community (eg, GPs, pharmacists and hospitals).

## Views about deprescribing and its benefits

Most participants, especially care home staff, reported having mostly positive views and experiences of deprescribing. Often, staff and residents articulated that if medicines were not needed or did not improve the resident's well-being then they would be happy to support deprescribing. Some residents expressed a few concerns because some medicines were perceived as indispensable. Residents discussed concerns about the consequences if these were deprescribed and how some medicines were necessary or too important (eg, antipsychotics for schizophrenia). Care home staff reported the various potential benefits to resident health, quality of life, costs, time and safety. All participants thought that a deprescribing process should be individualised to the resident and, therefore, adaptable, pilotable and reversible. These beliefs and attitudes of individuals seem to instil a dynamic, tailored, responsive deprescribing culture in the care home.

## Ability to question and contribute

A key characteristic discussed by care home staff and residents was the capacity and confidence to question or raise concerns about medicines with GPs or other prescribers. However, most care home staff and residents felt that they would default to GP advice. Care home staff stated how they did not have the training to question prescribing decisions, but they conveyed that questioning medicines was important to advocate for residents. For those staff who did feel confident to question or raise concerns, the expectation was that they would be listened to by the doctor. Care home staff recognised their knowledge about residents and wanted it to be valued in deprescribing conversations and decisions.

## Relationships and communication

Care home staff discussed how mutual trust and respect between care homes and healthcare staff enabled collaboration and good working relationships. The collaboration between GPs and care homes was considered a crucial determinant for deprescribing, despite the support of other healthcare professionals (eg, pharmacists and other specialists). Supporting the deprescribing process and this collaboration was a good working relationship and information sharing between primary care and the care home. Care homes relied on professionals to review

medications, deprescribe and provide information about monitoring following changes; while the GP relied on the feedback provided by the care home on resident health. The exchanges and information sharing between care home staff and GPs (and other healthcare professionals) determined the access to knowledge and evidence to facilitate deprescribing. This communication was considered an essential component of the deprescribing process. For example, care home staff, the GP and residents discussed how deprescribing could be hindered if medication changes were not communicated to the care home. Care home staff also discussed their role in keeping family members informed, which was described as more challenging when they were unable to visit homes during the pandemic.

Notably, the distinct types of care home providers (independently owned and part of a larger organisation) did not seem a crucial factor influencing deprescribing. More critical were the working relationships with GP practices and support at individual care homes (such as regular medication reviews or access to pharmacists). The local quality of the relationships and how they worked with healthcare professionals could vary greatly among the providers and individual care homes.

## DISCUSSION

The overall aim of this qualitative study was to identify factors which influence deprescribing in English care home settings. The major themes found related to how deprescribing was a complex process, with multiple steps and the involvement of multiple people with distinct roles and internal and external contextual factors, concerning the characteristics of staff and residents and communication and relationships with the wider healthcare community. Local context and available support were found to be crucial factors across types of care home providers.

The findings add to previous deprescribing research, which described the barriers and enablers[21–27 29]; this study described in detail the process of deprescribing and the contextual factors which influence it in the English care home setting. It also considered the various roles and relationships of these different individuals involved in deprescribing. The present study found that how these different people worked together was an important determinant of deprescribing. Participants discussed and stressed how important collaboration and working relationships between care homes and healthcare providers (eg, GPs and other prescribers) more than the organisational structure of the care home provider. This finding suggests that these roles and relationships need to be addressed in any successful deprescribing approach in care homes.

It must be recognised that there are power differentials and a 'hierarchy' (with GPs holding the decision-making power) in these relationships, and the management of care home residents' medicines has to take into consideration who has decision-making power.[30] Hierarchies and

power differentials can affect whose voice is heard; it is well documented that when power differentials between health and social care settings are present healthcare priorities invariably dominate.[39] Fostering collective purpose and identity across sectors could ensure communication and collaboration are not negatively affected.[40] Care home residents, family members, care home staff and relevant healthcare professionals should be involved in the deprescribing process (if possible and appropriate) and their respective knowledge recognised and valued for shared decision-making. Additional training, tools, support and opportunities may need to be made available to care home staff so they can feel confident and able to question or raise concerns when they think that something is not quite right with the medication. Known enablers of psychological safety and patient safety culture in healthcare teams (such as, professional responsibility, open communication, peer support change-oriented leadership and learning orientation) could be implemented.[41 42] Future research could examine these separate roles and working relationships as well as resources to support them in greater detail to determine what activities and strategies encourage appropriate deprescribing.

Previous research has recommended nurse champions deprescribing in long-term care facilities,[30] but UK care homes do not always have access to qualified nurses. A recent trial introduced pharmacists to lead deprescribing in UK care homes,[43] but similar to this study's findings, the role of the pharmacist was not always understood by care home staff. Any initiative will need to educate not only those individuals selected to champion deprescribing but also the rest of those involved so there is a shared understanding of each other's role and contribution. The involvement of multiple individuals and organisations also raises issues related to legal concerns, higher workloads and duty of care.[21] Successful integrated working requires trust between health professionals and a clear understanding of responsibilities.[44] Further work may need to explore how these are established and negotiated in the care home-primary care working relationships.

The findings support previous work and policy promoting the involvement of multidisciplinary teams[30] and collaboration across care settings[45] to facilitate and implement deprescribing safely and efficiently. Recent policy changes in England set out standards for primary care services to support the delivery of the 'Enhanced Health in Care Homes' framework.[46] A major component of enhanced primary care support includes routine structured medication reviews. This mandated support from primary care is not only an impetus for medicine reviews and potential deprescribing in care homes but also provides regular opportunities for planning, decision-making and reflecting on their practice. But, evidence suggests that the adoption of the Enhanced Health in Care Homes Framework is not universal or uniform.[47] Accordingly, not all care homes will have access to the same support for deprescribing so approaches may need to be tailored to the availability of local resources and,

as this study found, the resources and relationships for individual care homes were key factors influencing deprescribing. The findings contribute to the medicine optimisation national priority[18 48 49] and NICE recommendations for deprescribing in care homes.[17 18] With an increasing ageing population and greater demand for care in the community, medicine optimisation will continue to be a priority for healthcare services.

### Strengths and limitations

A strength of this work is the use of a comprehensive, well-recognised implementation science framework, CFIR, to investigate how to implement deprescribing in real-world settings,[20 50] specifically in care homes.[51] The project demonstrates how CFIR can be used to identify the factors influencing deprescribing into practice.[32 33] The present study findings denoted several domains and constructs of CFIR and the relationships between these determinants. Using this framework can identify crucial determinants, address barriers to deprescribing and help ensure that interventions are effective and sustainable.[33 52] Another strength of the study was the number of respondents that participated in the context and conditions of the pandemic and the high demand for healthcare services where care staff have been so overwhelmed by pressures. Finally, this research adopted a strong PPI and partnership approach, which contributed greatly to the delivery and success of the study.

Some limitations of the study must be acknowledged. Due to the COVID-19 pandemic, care home observations and focus groups were not conducted, which had been originally planned[34]; this could have provided a better understanding of everyday practices and insight into experiences than interviews alone. Future work could adopt an ethnographic approach to explore the culture of deprescribing in care homes and observe deprescribing conversations and behaviours. By only conducting interviews, we had to exclude participants who are unable to converse in English and show signs of severe cognitive impairment. These individuals may have different experiences and perspectives. Interviews were with family and friends of these older adults to try to capture these experiences and views. Other limitations of the study were that two-thirds of the sample were care home staff (especially, those in senior positions) and most of the participants were female and white so the findings reflect mostly their perspectives and experiences. Moreover, there may be bias in the sample, as regional managers and directors were involved in the selection of participants. Participants were informed that their responses would be kept confidential, and their employment or care (or family member's care) would not be affected by their responses; they did provide both positive and negative experiences. There was only one prescriber interviewed, but as we were principally interested in how deprescribing would work in a care home setting, the recruitment of care home staff, residents and families were prioritised. Most

previous research has studied the views of prescribers[22–24] and rarely included the views of care home staff.

From the participants' responses in this study, a difference in views between provider types was not noted but it does not mean there one is not present. Further research could explore the views of more providers and care homes to explore possible differences. Involving care homes in research is challenging and there are several barriers to research participation.[53] The COVID-19 pandemic has shown how crucial care homes and their staff are in the care of the frailest and vulnerable. The skills, dedication, and compassion of care home staff not only shape care but can greatly contribute to research.

## Conclusion

For deprescribing to be successfully undertaken in care homes, several influencing factors (such as individuals' beliefs and abilities, relationships and communication) need to be considered and the process itself of deprescribing (including preparation, engaging stakeholders and monitoring). Additional training, tools, support and opportunities need to be made available to care home staff, so they can feel confident and able to question or raise concerns about medicines with prescribers. Further work is warranted to design and implement a deprescribing approach for care homes that can be achieved within current structures and resources.

**Author affiliations**
[1]Centre for Research in Public Health and Community Care, University of Hertfordshire, Hatfield, UK
[2]National Institute for Health Research Applied Research Collaboration East of England, Cambridge, UK
[3]Department of Global Health & Social Medicine, King's College London, London, UK
[4]University of Exeter Medical School, University of Exeter, Exeter, UK
[5]National Institute for Health Research Applied Research Collaboration South West Peninsula, Exeter, UK
[6]Patient and Public Involvement Group, National Institute for Health Research Applied Research Collaboration South West Peninsula, Exeter, UK
[7]Somerset NHS Foundation Trust, Taunton, UK
[8]Classic Care (Devon) Homes Ltd, Teignmouth, UK
[9]Somerset Care Ltd, Taunton, UK

**Acknowledgements** We appreciate the feedback from care home staff and residents who attended the patient and public involvement events and who took part in the study. We gratefully acknowledge the input of Donald Nigel Reed. The South West Clinical Research Network community support team collaborated and supported the research team.

**Contributors** KW led the planning, conduct and reporting of the work in this manuscript, and is the guarantor of the overall content. KW, JD, EC, GC, LP and KS supported the conceptualisation of the work and acquisition of the funding. TC and GC supported the recruitment of participants and data collection. JR, JD and AA contributed to the analysis of the data. LP and BW reviewed for important intellectual content. All authors have reviewed and approved the final version of the paper.

**Funding** This project is funded by the National Institute for Health Research (Research for Patient Benefit, Understanding stakeholders' perspectives on implementing deprescribing in care homes (STOPPING), PB-PG-0418-20026). This is independent research supported by the National Institute for Health Research (NIHR) Applied Research Collaboration for the East of England and South West Peninsula. The views expressed in this publication are those of the authors and not necessarily those of the NHS, NIHR or the Department of Health and Social Care.

**Competing interests** None declared.

**Patient and public involvement** Patients and/or the public were involved in the design, or conduct, or reporting, or dissemination plans of this research. Refer to the Methods section for further details.

**Patient consent for publication** Consent obtained directly from patient(s).

**Ethics approval** This study involves human participants. Ethical approval to conduct the study was given by the Social Care Research Ethics Committee (19/IEC08/0058). Participants were provided with an information sheet stating the goals of the study and required to sign a consent form to indicate their willingness to participate. Voluntary participation and the right to ask any questions and to decline participation at any time will be emphasised during the data collection. Participants gave informed consent to participate in the study before taking part.

**Provenance and peer review** Not commissioned; externally peer reviewed.

**Data availability statement** Data are available upon reasonable request. The datasets used and/or analysed during the current study are available from the corresponding author on reasonable request.

**ORCID iDs**
Krystal Warmoth http://orcid.org/0000-0003-0615-5778
Jessica Rees http://orcid.org/0000-0002-9471-2134

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
