## [Reviewer comments · BMJ Open]

ARTICLE DETAILS

TITLE (PROVISIONAL)	DETERMINANTS OF IMPLEMENTING DEPRESCRIBING FOR OLDER ADULTS IN ENGLISH CARE HOMES: A QUALITATIVE INTERVIEW STUDY
AUTHORS	Warmoth, Krystal; Rees, Jessica; Day, Jo; Cockcroft, Emma; Aylward, Alex; Pollock, Lucy; Coxon, George; Craig, Trudy; Walton, Bridget; Stein, Ken

VERSION 1 – REVIEW

REVIEWER	Potter, Kathleen Ryman Healthcare Ltd
REVIEW RETURNED	29-Jul-2023

GENERAL COMMENTS	This paper describes a qualitative study designed to identify factors influencing deprescribing activity in older adults living in care homes in England. The investigators recruited and interviewed 36 people for the study, of which the majority were care home staff(23) and only 1 was an actual prescriber (one GP). The participants were recruited from 15 different care homes, representing two different care provider organisations, one large and one small. The authors are to be commended on undertaking a rigorous and well-designed project to investigate attitudes to and barriers to deprescribing in people living and working in care homes in the UK. It is always difficult to critique the methods of qualitative studies, as they are very much descriptive in nature and as long as the authors report that they followed the guidelines and tried to avoid inserting their own opinions, it is impossible to know whether this is actually true or not. I have a number of general criticisms of the paper. First, several of the references cited are not from peer-reviewed publications and make claims that need to be verified from the actual scientific literature rather than quoting from government summaries, which can be a little loose with the facts or written with a specific agenda. I would feel more confident in the claims made by the authors if they cited original studies. My concern is that claims often get exaggerated far beyond the original evidence when quoted down a trail of citations that simply cite other citations. For example, the authors reference a claim in Line 51 (reference 11) that “deprescribing can mitigate these harms “(I’m assuming they are referring to the harms of polypharmacy, as it is not entirely clear from the sentence). However the reference they quote is a “Systemic review of the emerging definition of deprescribing”. This is not a randomised control trial showing that deprescribing actually reduces harm. It is a review of the literature
--

	designed to find a definition of deprescribing. This somewhat reduces my confidence that the authors have actually read the literature they are citing. Secondly, the recruitment methods may have led to an extremely biased sample, not representative of the majority of either carers or residents. As far as I could tell, there was no attempt to recruit randomly, the home sites were selected by care staff in collaboration with the researchers and participating care staff were selected by managers of the care home organisations. Similarly, residents and family members were selected by care home staff. These selection methods ring loud alarm bells for me. I would have thought some attempt to sample randomly would have provided a much better cross-section of opinion than hand-picking participants. I was also disappointed to see the inclusion of one single prescriber in the study cohort. Most of the people interviewed identified the GP as the key decision-maker about whether medicines were continued or ceased. Why were more GPs not recruited by the investigators once this information became available? Any practical strategy to reduce polypharmacy in residential care is going to have to include the GP, whether everyone else likes it or not. While this may be frustrating for nurses and care staff, it is reality of the current hierarchical system in healthcare where prescribing decisions are made by doctors in consultation with their patients. Changing the system may be an admirable goal (or not) but if the authors genuinely want to reduce the harms of polypharmacy for the people who are suffering from it now, it would pay to work with the system that exists rather than fantasizing about revolutionising it. Which means including GPs in any research designed to investigate the difficulties of deprescribing. You are probably picking up a tone of frustration in this review. I am frustrated. I have read so many of these studies “investigating the barriers to deprescribing”. The only conclusion I can come to is that these studies are much easier to undertake than actually generating the evidence to support deprescribing. Which is the actual thing that is required to change practice and is still very sparse. The biggest barrier to deprescribing is the lack of evidence to support it. If you generate the randomised evidence that it is a “good thing”, then prescribers will deprescribe. But currently, despite all the platitudes about “evidence that it can reduce the harms of polypharmacy”, the evidence is still very poor, of low quality, or just simply absent. So while the authors have done an admirable job on this research and carefully followed all the guidelines for producing a high quality qualitative study, I think ultimately it was an absolute waste of their time and merely describes what is as obvious as a nose on a face to anyone involved in providing care to people living in care homes – ie. deprescribing is a complex undertaking, with many people involved with many different agendas, and until prescribers stop creating polypharmacy in response to the pressures to do so, very little is going to change.
--	---

REVIEWER	Olesen, Anne Mech-Sense, Department of Gastroenterology and Hepatology, Aalborg University Hospital
-----------------	---

GENERAL COMMENTS

The study used qualitative semi-structured interviews to explore the factors that may help or hinder deprescribing practice for older people within care homes. The research was carried out effectively, and the manuscript is both comprehensible and skilfully composed. Nonetheless, I do have a couple of minor remarks.

Abstract

Part of the conclusion “deprescribing implemented safely and successfully in care homes can benefit residents”. This cannot be concluded from the findings of the study as clinical outcomes and patient safety was not assessed. Please revise the conclusion so that it only concludes on study results.

Results

The variable “hours worked per week” could preferably be dichotomized to 1) more than 32 hours and 2) 32 hours or less, as low numbers are reported in each category.

Specification of care home staff is missing. As different countries employ different care home staffs, it is essential to describe the specific roles of the care home staff included in the study. E.g., how were they included in the medication process? Were they educated in medication or only care? Were all included care home staff members similar in education? “Care home staff” may not be considered a single group of staff members.

In contrast “education” information on residents and carer seems irrelevant. Please argue why this is relevant and add to the discussion if you decide to keep information.

Discussion

Psychological safety and patient safety culture could be discussed in page 14 line 309-314, as highly relevant issues.

Further discussion is needed regarding relationships and hierarchy. How is this secured, is it possible in all countries or maybe more difficult in some countries than in others? Hierarchy and differences between countries could be discussed for an international interest.

Which other factors may affect hierarchy? age of GP? age of care home staff? experiences? private relations? leadership?

It could be discussed how the selection of care home staff, by regional managers or directors may have affected outcomes.

Did the selection of care homes affect relationship? maybe care homes with collaborative challenges were not selected? This could be discussed too.

In the discussion it is stated that “The present study found that how these different people worked together was an important determinant of deprescribing, even more than the differences in care home provider organizational structure” which findings supports this statement of “even more”. How was “how people worked together” assessed?

In the discussion it is stated that “A ‘hierarchy’ (with GPs holding the decision-making power) of the management of care home residents’ medicines and care may need to be addressed so communication and collaboration are not negatively affected”.

Further discussion is needed. How is this secured, and is it possible in all countries or maybe more difficult in some countries than in others? Which factors may affect hierarchy? age of GP? age of care home staff? experiences? private relations? leadership?

	Conclusion Regarding the phrase “Deprescribing implemented safely and successfully in care homes can benefit residents”, this cannot be concluded from the findings of the study. Please revise the conclusion so that it only concludes on objectives and study results.
--	--

VERSION 1 – AUTHOR RESPONSE

Response to Reviewers

For ease of reading, we have numbered all comments from Reviewers, chronologically, and have italicised the comments and bolded the start of each of our Responses.

Reviewer 1 comments

Comment #1: The authors are to be commended on undertaking a rigorous and well-designed project to investigate attitudes to and barriers to deprescribing in people living and working in care homes in the UK. It is always difficult to critique the methods of qualitative studies, as they are very much descriptive in nature and as long as the authors report that they followed the guidelines and tried to avoid inserting their own opinions, it is impossible to know whether this is actually true or not.

Response: We thank the Reviewer for their commendation of the methods of our research. However, we do not agree with their comment that qualitative research in itself is flawed. Some research questions cannot be answered using (only) quantitative methods. Such as this study which was exploring the different factors that may help or hinder deprescribing practice for older people within care homes, this exploration would not have been feasible using quantitative methods. Moreover, we disagree with the suggestion that qualitative research findings can be false or misleading because of its descriptive or biased nature. We have taken great efforts in this work to minimize bias and demonstrate rigour (which they have commended us on).

Comment #2: First, several of the references cited are not from peer-reviewed publications and make claims that need to be verified from the actual scientific literature rather than quoting from government summaries, which can be a little loose with the facts or written with a specific agenda. I would feel more confident in the claims made by the authors if they cited original studies. My concern is that claims often get exaggerated far beyond the original evidence when quoted down a trail of citations that simply cite other citations. For example, the authors reference a claim in Line 51 (reference 11) that “deprescribing can mitigate these harms “(I’m assuming they are referring to the harms of polypharmacy, as it is not entirely clear from the sentence). However the reference they quote is a “Systemic review of the emerging definition of deprescribing”. This is not a randomised control trial showing that deprescribing actually reduces harm. It is a review of the literature designed to find a definition of deprescribing. This somewhat reduces my confidence that the authors have actually read the literature they are citing.

Response: We have revised the manuscript including scientific evidence, where appropriate or available, for the claims included. We are sorry for this misunderstanding with the reference and will revise the sentence, mentioned by the Reviewer in the Introduction, so the reference is to how deprescribing is defined and include references to evidence about deprescribing mitigating the harms of polypharmacy.

Comment #3: Secondly, the recruitment methods may have led to an extremely biased sample, not representative of the majority of either carers or residents. As far as I could tell, there was no attempt

to recruit randomly, the home sites were selected by care staff in collaboration with the researchers and participating care staff were selected by managers of the care home organisations. Similarly, residents and family members were selected by care home staff. These selection methods ring loud alarm bells for me. I would have thought some attempt to sample randomly would have provided a much better cross-section of opinion than hand-picking participants.

Response: The sample was recruited purposefully, not randomly, as is common practice in qualitative research to focus in depth on the topic. We wanted to gain a better understanding of deprescribing in care homes so we specifically recruited those with experience of polypharmacy and deprescribing. Previous research captured prescribers but care home staff views are rarely included (see lines 81-87 and 396-400). Residents and their family carers were selected using our criteria (i.e. experience of polypharmacy and deprescribing, absence of serious cognitive impairment and ability to consent) and approached by the managers to take part in the study. We understand the sample of residents and family carers is limited because of this method and have discussed the weaknesses of the resident and carer recruitment and their sample size in a revised manuscript (see Discussion, Strengths and limitations section).

Comment #4: I was also disappointed to see the inclusion of one single prescriber in the study cohort. Most of the people interviewed identified the GP as the key decision-maker about whether medicines were continued or ceased. Why were more GPs not recruited by the investigators once this information became available?

Response: We agree that unfortunately this does affect our findings and have noted the sample as a limitation to the study in the Discussion. We have further discussed this limitation (only one prescriber) in the revised manuscript.

Comment #5: Any practical strategy to reduce polypharmacy in residential care is going to have to include the GP, whether everyone else likes it or not. While this may be frustrating for nurses and care staff, it is reality of the current hierarchical system in healthcare where prescribing decisions are made by doctors in consultation with their patients. Changing the system may be an admirable goal (or not) but if the authors genuinely want to reduce the harms of polypharmacy for the people who are suffering from it now, it would pay to work with the system that exists rather than fantasizing about revolutionising it. Which means including GPs in any research designed to investigate the difficulties of deprescribing.

Response: We want to clarify that we did not state that deprescribing would not involve a GP or other prescriber. Our finding was that deprescribing involves multiple people working together and they must be considered for any successful deprescribing approach in care homes. We did not advocate to 'revolutionising' the system. Instead, we suggested the potential opportunities for planning, decision-making, and reflecting on their practice, such as multidisciplinary teams and routine medication reviews, which are in place due the adoption of the Enhanced Health in Care Homes Framework in England (see Discussion, page 15).

Comment #6: You are probably picking up a tone of frustration in this review. I am frustrated. I have read so many of these studies "investigating the barriers to deprescribing". The only conclusion I can come to is that these studies are much easier to undertake than actually generating the evidence to support deprescribing. Which is the actual thing that is required to change practice and is still very sparse. The biggest barrier to deprescribing is the lack of evidence to support it. If you generate the randomised evidence that it is a "good thing", then prescribers will deprescribe. But currently, despite all the platitudes about "evidence that it can reduce the harms of polypharmacy", the evidence is still very poor, of low quality, or just simply absent.

Response: We are sorry that Reviewer 1 was frustrated after reading the manuscript and similar studies. We would argue that not just 'randomised evidence' will make prescribers deprescribe. There

are evidence-based guidelines that recommend deprescribing and medicines optimisation. For example, National Institute for Health and Care Excellence (NICE) recommends deprescribing as part of the comprehensive medication review of a person with multiple long-term conditions (references 12-14). Therefore, there is a need to consider the context and perceptions of deprescribing for the translation of these recommendations. In the manuscript, we discuss an evidence summary which of deprescribing in long-term care facilities from nine different countries (reference 26), which found social influences and environmental factors were the key barriers and enablers to deprescribing. We explored these factors using of a comprehensive, well-recognised implementation science framework; an approach has been advised previously (references 28 & 30). By using this framework, we gained a better understanding of the possible determinants. We investigated how to implement deprescribing in real-world settings. Specifically, we focused on care homes, where polypharmacy, the risk of medication errors, and complications due to frailty are high.

Comment #7: So while the authors have done an admirable job on this research and carefully followed all the guidelines for producing a high quality qualitative study, I think ultimately it was an absolute waste of their time and merely describes what is as obvious as a nose on a face to anyone involved in providing care to people living in care homes – ie. deprescribing is a complex undertaking, with many people involved with many different agendas, and until prescribers stop creating polypharmacy in response to the pressures to do so, very little is going to change.

Response: We thank the Reviewer for their positive comments, but we strongly disagree with their statement that the research ‘was an absolute waste of [our] time and merely describes what is as obvious as a nose on a face to anyone involved in providing care to people living in care homes’. The findings may have been obvious to them (or someone working in care homes), whilst those working in a similar field or those with personal/anecdotal evidence about care homes may feel they 'know' the situation, we stand by the evidence conducted/analysed and presented in this paper as providing a valuable and new contribution.

Reviewer 2 comments

Comment #1: The study used qualitative semi-structured interviews to explore the factors that may help or hinder deprescribing practice for older people within care homes. The research was carried out effectively, and the manuscript is both comprehensible and skilfully composed. Nonetheless, I do have a couple of minor remarks.

Response: We thank Reviewer 2 for their positive comments about the study and writing.

Comment #2: Part of the conclusion “deprescribing implemented safely and successfully in care homes can benefit residents”. This cannot be concluded from the findings of the study as clinical outcomes and patient safety was not assessed. Please revise the conclusion so that it only concludes on study results.

Response: We have revised the Abstract Conclusions so that it only includes study results.

Comment #3: The variable “hours worked per week” could preferably be dichotomized to 1) more than 32 hours and 2) 32 hours or less, as low numbers are reported in each category.

Response: We have made this change in the reporting in Table 1.

Comment #4: Specification of care home staff is missing. As different countries employ different care home staffs, it is essential to describe the specific roles of the care home staff included in the study. E.g., how were they included in the medication process? Were they educated in medication or only care? Were all included care home staff members similar in education? “Care home staff” may not be considered a single group of staff members.

Response: We thank the Reviewer for bringing this to our attention, and we agree that there are different roles for care home staff. We have provided more information about the tasks and responsibilities of those staff in Table 1.

Comment #5: In contrast "education" information on residents and carer seems irrelevant. Please argue why this is relevant and add to the discussion if you decide to keep information.

Response: We removed this information in the table.

Comment #6: Psychological safety and patient safety culture could be discussed in page 14 line 309-314, as highly relevant issues.

Response: We thank the Reviewer for this suggestion and believe it is relevant. We have added literature on these issues this to a revised manuscript in the Discussion.

Comment #7: Further discussion is needed regarding relationships and hierarchy. How is this secured, is it possible in all countries or maybe more difficult in some countries than in others? Hierarchy and differences between countries could be discussed for an international interest. Which other factors may affect hierarchy? age of GP? age of care home staff? experiences? private relations? leadership?

Response: We have added more discussion about the relationships and hierarchy in the Discussion; however, we cannot comment on other countries or what factors affect hierarchy, age, experience, etc. as it is out of the scope of this qualitative work.

Comment #8: It could be discussed how the selection of care home staff, by regional managers or directors may have affected outcomes.

Response: We have discussed this limitation in the revised manuscript.

Comment #9: Did the selection of care homes affect relationship? maybe care homes with collaborative challenges were not selected? This could be discussed too.

Response: We cannot comment on how and if the relationships were affected by the selection of care homes. This is outside of the scope of the study.

Comment #10: In the discussion it is stated that "The present study found that how these different people worked together was an important determinant of deprescribing, even more than the differences in care home provider organizational structure" which findings supports this statement of "even more". How was "how people worked together" assessed?

Response: We have clarified this statement in a revised manuscript. We cannot comment on the assessment of "how people worked together" as this was outside the scope of the study.

Comment #11: In the discussion it is stated that "A 'hierarchy' (with GPs holding the decision-making power) of the management of care home residents' medicines and care may need to be addressed so communication and collaboration are not negatively affected". Further discussion is needed. How is this secured, and is it possible in all countries or maybe more difficult in some countries than in others? Which factors may affect hierarchy? age of GP? age of care home staff? experiences? private relations? leadership?

Response: See response to Reviewer 2 comment #7.

Comment #12: Regarding the phrase "Deprescribing implemented safely and successfully in care homes can benefit residents", this cannot be concluded from the findings of the study. Please revise the conclusion so that it only concludes on objectives and study results.

Response: We removed statement in revised manuscript so it more aligns with the study findings.